# Monitoring Eurasian Woodcock (*Scolopax rusticola*) with Pointing Dogs in Italy to Inform Evidence-Based Management of a Migratory Game Species

Marco Tuti [1], Tiago M. Rodrigues [2], Paolo Bongi [3,*], Kilian J. Murphy [4], Paolo Pennacchini [5], Vito Mazzarone [6] and Clara Sargentini [1]

1   Dipartimento di Scienze e Tecnologie Agrarie, Alimentari, Ambientali e Forestali (DAGRI), University of Florence, Via delle Cascine, 5, I-50144 Firenze, Italy

2   Divisão de Caça Pesca e Parques Direção Regional dos Recursos Florestais Rua do Contador 23, 9500-050 Ponta Delgada Açores, Portugal

3   Hunting Office ATC MS13, L.go Bonfigli 3/5, I-54011 Aulla, Italy

4   School of Biology and Environmental Science, University College Dublin, Science Centre—West, Belfield, D04 V1W8 Dublin, Ireland; kilian.murphy@ucdconnect.ie

5   FANBPO, "Fédération des Associations Nationales des Bécassiersdu Paléarctique Occidental" (President) and FIBec, "Federazione Italiana Beccacciai" (President), Via Fausto Vagnetti 12, I-52031 Anghiari, Italy

6   Tuscany Region, Administral Regional Hunting Office, Via di Novoli 26, I-50127 Firenze, Italy

*   Correspondence: paolo.bongi@atcms13.it

**Abstract:** The phenology of migratory bird species is a crucial aspect of their biology that has far-reaching implications for wildlife management, particularly when these species are hunted as game. For this reason, many monitoring projects have investigated the presence of Western European bird species in diverse Palearctic regions using abundance indexes. Here, our aim was to define Woodcock's presence in Italy during the post-nuptial migration, the wintering phase, and at the beginning of the pre-nuptial migration phase, using monitoring data collected between September and March for the period 2016 to 2021. The presence of Woodcock in Italy and other regions of the Mediterranean basin can be compared using an index, specifically the "Indice Cynégétique d'Abondance" (ICA) which corresponds to the number of different Woodcock flushed during a hunting trip. We modelled the abundance of Woodcock as a function of biotic (habitat type, vegetation) and abiotic (place, season, temperature, altitude) factors to assess the presence of Woodcock in Italy Our findings reveal that temperature and altitude have an inverse effect on the abundance index of Woodcock in Italy, while deciduous woodland is a preferred habitat for the species. We observe an increase in Woodcock's presence from the end of September to late November, followed by a decrease in late January. Moreover, we have identified a significant rise in the ICA index during the latter part of February and early March, indicating the pre-nuptial migration period. Our study contributes significantly to our understanding of Woodcock migration phenology, particularly with respect to the management of the species in Italy and other Mediterranean basin states. Our results underscore the importance of long-term monitoring programs for evaluating key spatial population metrics such as presence and abundance, which are critical for sustainable hunting and effective conservation management of game species.

**Keywords:** autumn migration; wintering; pre-nuptial migration; Mediterranean area; abundance indices; monitoring

## 1. Introduction

To ensure adequate management plans and hunting quotas, it is imperative to manage game bird species empirically. This involves the use of current knowledge on population abundance, as many avian species fall under this category and require up-to-date information for effective management. With respect to migratory species, there is a lack of

consensus on which jurisdiction is responsible for monitoring, and importantly, when and where the monitoring should be prioritised. Often, when the breeding areas are in remote places, it is difficult to robustly implement monitoring programmes [1].

The Eurasian Woodcock *Scolopax rusticola* (hereafter Woodcock) is a migratory wader, widely distributed in the Palearctic as a breeding species [1,2]. The Woodcocks breeding range is extensive across this region and is inclusive of Scandinavia to Eastern Russia [3–6]. In Western Europe, the distribution of Woodcock is more fragmented and includes Ireland, the United Kingdom, Germany, France, the Alps, and Pyrenees Regions; as well, some sedentary populations inhabit northern Spain, the Azores, Madeira and Canary Islands [7].

The Mediterranean basin is an important wintering area for Woodcock but, until now this region has lacked a monitoring programme that can be replicated across the region. Monitoring programs aimed at quantifying Woodcock abundance have been developed in the UK [3], Spain [8,9], France [10–12], or Portugal [13]. In Italy, studies have largely focused on other aspects of this species biology, namely age and sex ratios [13–16], winter mortality [17], woodcock winter diet [18], genetics structure of wintering population [19,20], and some aspects of migratory behaviour but more research into this aspect of the species ecology is certainly needed [21].

Currently, the conservation status of Woodcock is classified as "Least Concern" on the red list of the International Union for the Conservation of Nature [22]. In Europe, the breeding trend appears stable [23]; the species is classified as "huntable" in the second annex of the EU directive 09/147, and, according to the most recent data, the European population is estimated between 15 and 20 million individuals [24,25].

For the majority of bird species, population abundance is investigated with direct observation of birds and calls associated with demonstrative breeding behaviours [26]. In wintering areas monitoring programs on Woodcock's abundance typically used direct observations to estimate population and quantified these estimates with abundance indices [8,13]. In estimating the abundance of game birds, various indices have been proposed, including those based on data collected by hunters during the hunting season [10], as well as those derived from nocturnal banding visits [10,27].

Abundance indices are commonly employed in ecology and are incorporated into monitoring programs due to their simplicity in calculation, ease of interpretation, and extensive history of application. Moreover, abundance indices that measure population abundance as the mean population size at occupied sites are generally preferred over occurrence indices, which are typically scaled indices that compare the proportion of sites in a defined region to the number of birds observed [28]. Numerous studies have utilized abundance indices to explore the population densities of game bird communities and have shed light on their spatial distribution, habitat preferences [29–31], species richness, as well as the impact of local environmental changes on their population dynamics [32,33]. Woodcock monitoring is therefore paramount to allow researchers to determine population abundance and correlate it with climatic, environmental, and anthropogenic changes to further inform the evidence-based management of the species.

We analysed systematic monitoring data collected in Italy, by Woodcock hunters, between September 2016 and March 2021, in areas where Woodcock hunting was permitted.

The primary goal of this study was to investigate the length of Woodcock post-nuptial migration, wintering period, and pre-nuptial migration phenology, by examining the trend of its abundance from the beginning of the autumn migration (first half of September) until the end of March, which can accurately represent the last period of stay in the Mediterranean region. This study constitutes the first step in creating national-scale monitoring of game birds in Italy, using the Woodcock as a demonstrative case study.

## 2. Methods

### 2.1. Field Survey

We developed a monitoring program in collaboration with Woodcock hunters, from September to March, 2016–2017 to 2020–2021, in Italy. The hunters who partook in the

study were trained through qualification courses, the primary objective of the qualification courses attended by the hunters involved in this study was to teach correct behavioural protocols for monitoring sessions, such as immediately blocking dogs upon sighting a Woodcock, to minimize disruptions to both target and non-target species. These courses were officially recognized and validated at the ministerial level to ensure that participants had the necessary competencies for monitoring. Following this training course, the participants were provided with practical demonstrations of data collection methods and were taught how to enter the collected data into the appropriate databases. Additionally, a practical test was administered to ensure that the participants fully understood the learning objectives. As a result of this training, we were confident that the observers possessed the necessary skills to accurately count and identify Woodcock. These data were submitted to an online database, named Beccapp (http://www.beccapp.it accessed on 15 April 2021), where each hunter registers himself using a personal username and password.

In each season we considered two periods: the hunting period (September to January) and the post-hunting period (February and March). In Italy, Woodcock hunting is allowed from the third Sunday of September to the 31 of January, with some differences between regions in starting and ending of the hunting season. In our analyses, we took into account the appropriate start and end dates of the hunting season, and we used the terms "hunting trip" and "monitoring trip" to distinguish between data collection trips conducted during the hunting period and those conducted during the post-hunting period. During these periods, the surveyors reported the following information for each hunting or monitoring trip: date, place (at Municipality scale), time spent in hunting or monitoring (hours and minutes), number of Woodcock flushed (their estimate of different birds), number of participants, number of pointing dogs used, temperature (recorded with a personal smartphone), altitude (recorded with a personal smartphone, for altitude bands of 500 m), and vegetation type using three categories: deciduous wood, coniferous wood, brushland.

Our monitoring protocol recognized the potential for double counting of Woodcock that had been flushed and subsequently relocated. To mitigate this risk, we provided training to surveyors and instructed them to count only those Woodcock that flew in the opposite direction from the observer after being flushed. The information collected allowed us to estimate the abundance of Woodcock in the form of a hunting index of abundance (ICA—"Indice Cynégétique d'Abondance"; [12]), which corresponds to the number of different Woodcock flushed during a hunting trip, considering a standard duration of 3.5 h, divided for the effective duration of the trip. A standard duration of 3.5 h, is a parameter adopted by Fadat [12] and it represents the mean duration of a Woodcock hunting trip in Europe. Two other important parameters have been used in the calculation of ICA: the number of hunters and the number of pointing dogs involved in monitoring sessions [13].

Specifically, ICA is calculated as:

ICA: (number of different Woodcock flushed/trip duration/number of participants/ number of pointing dogs used) * 3.5.

During the hunting period, according to Italian national law n° 157/92, the hunting trips could be carried out five days weekly (excluding Tuesdays and Fridays). During the post-hunting period, which began on the 1st of February or on the day that Woodcock hunting closed, and continued until the 31st of March, monitoring trips were conducted on four fixed days each week (Tuesday, Friday, Saturday, and Sunday). In both the hunting and post-hunting periods, the monitoring trips focused on investigating huntable zones, which were predominantly characterized by wooded areas. Due to the differences in data between the hunting and monitoring periods, with shooting being permitted during the hunting period but not during the monitoring period (resulting in no removal of birds and likely less disturbance), separate analyses were conducted for each period.

*2.2. Statistical Analysis*

We built general linear models (GLM) with annual Woodcock ICA as a response variable, fitted to a Gaussian error structure, and using place (Municipality's latitude) as a factor, and habitat type as a covariate.

We conducted the analyses considering the municipalities always present in the monitoring seasons, as well as taking into account all the municipalities present over the five-year period.

To assess the impact of temperature, altitude, and vegetation type on the abundance of Woodcock, we used three separate zero-inflated negative binomial (ZINB) regression models. In each model, the number of flushed Woodcock was the dependent variable, and the logarithm of trip duration was used as an offset. However, since these variables were not available for the entire dataset, we were unable to incorporate them into subsequent analyses. The variation in Woodcock abundance was evaluated during and between seasons, with generalized additive mixed models (GAMM), with a negative binomial error family. We included in the models season, the day of the season as a "cycling" variable, coordinates of the centroid of where the hunting trip took place, the random effect of the hunter, and the logarithm of the duration of the trip divided by the number of dogs and by the number of hunters participating in that trip, was included as an offset (to accommodate differences in sampling effort). We developed three models, full season (hunting period + post-hunting period), hunting period, and post-hunting period, and we reported a confidence interval of 95% for variation of Woodcock abundance.

All statistical analyses were performed in R 4.0.5 [34] and the significance of the test was considered at $p < 0.05$.

## 3. Results

We used a total of 36,443 trips registered in the Beccapp for our analysis, produced by 828 hunters, and a total of 2097-pointing dogs, with an average of 28.9 hunter/trip and 53.6-pointing dogs/trip. During the hunting period from 2016–2017 to 2020–2021 (September-January of each hunting season) we obtained 29,368 Woodcock hunting trips, with pointing dogs, while during the post-hunting period (in February and March) we obtained 7075 Woodcock monitoring trips, with pointing dogs (no shooting), during the post-hunting period (Table 1).

During these trips, over 131,590 h were spent searching for Woodcock, in 1588 of the 7998 Italian communes (Figure 1), resulting in 55,467 contacts with individual birds.

In preliminary analyses, we modelled the full dataset, and there was no decrease in abundance, instead, the abundance even increased during the monitoring period.

The abundance of Woodcock increased with temperature (ZINB, $p < 0.001$; Figure 2a), decreased with altitude (ZINB, $p < 0.001$; Figure 2b), and varied by habitat type (ZINB, $p < 0.001$; Figure 2c), with coniferous woodland showing the lowest presence of Woodcock and deciduous woodland having the highest presence of the species.

The results of the full season model (hunting period + post-hunting period) were nearly identical to those obtained with data limited to the hunting period (Table 2). The model exhibited some over-dispersion and explained 37.3% of the variance (1.36). Hunter, location, and day of the hunting period (Table 2) significantly affected the amount of Woodcock flushed. Furthermore, these impacts appeared to be more relevant than seasonal variation, which was equally significant (Table 2).

The Woodcock encounter, and so the ICA values, showed a difference in relation to geographic information, recorded as coordinates (x, y) of municipalities where monitoring trips were carried out (Table 2).

The total abundance of Woodcock decreased between the 2016–2017 and 2017–2018 seasons, but after a significant recovery, in 2018–2019, remained stable until 2020–2021 (Figure 3). For instance, the decrease from 1.0 in 2016–2017 to ~0.8 in 2017–2018 roughly corresponds to a decrease of 20%.

**Table 1.** Number of hunting/monitoring trips registered per decade, per season.

| Sampling | Month | Decade | Season 2016–2017 | 2017–2018 | 2018–2019 | 2019–2020 | 2020–2021 |
|---|---|---|---|---|---|---|---|
| Hunting period | September | -2 | 2 | 7 | 1 | 5 | 1 |
| | | 3 | 20 | 16 | 17 | 13 | 14 |
| | October | 4 | 71 | 81 | 109 | 150 | 141 |
| | | 5 | 117 | 144 | 272 | 448 | 342 |
| | | 6 | 278 | 436 | 438 | 729 | 766 |
| | November | 7 | 291 | 555 | 746 | 983 | 932 |
| | | 8 | 416 | 635 | 947 | 773 | 595 |
| | | 9 | 312 | 416 | 831 | 926 | 571 |
| | December | 10 | 339 | 489 | 1016 | 964 | 580 |
| | | 11 | 272 | 321 | 676 | 770 | 1093 |
| | | 12 | 281 | 381 | 833 | 894 | 399 |
| | January | 13 | 166 | 220 | 482 | 665 | 478 |
| | | 14 | 169 | 261 | 386 | 608 | 574 |
| | | 15 | 209 | 205 | 251 | 371 | 466 |
| Post-hunting period | February | 16 | 33 | 38 | 170 | 205 | 174 |
| | | 17 | 142 | 128 | 269 | 296 | 466 |
| | | 18 | 96 | 164 | 271 | 468 | 582 |
| | March | 19 | 110 | 245 | 546 | 439 | 476 |
| | | 20 | 170 | 243 | 230 | 13 | 433 |
| | | 21 | 89 | 176 | 225 | 4 | 174 |

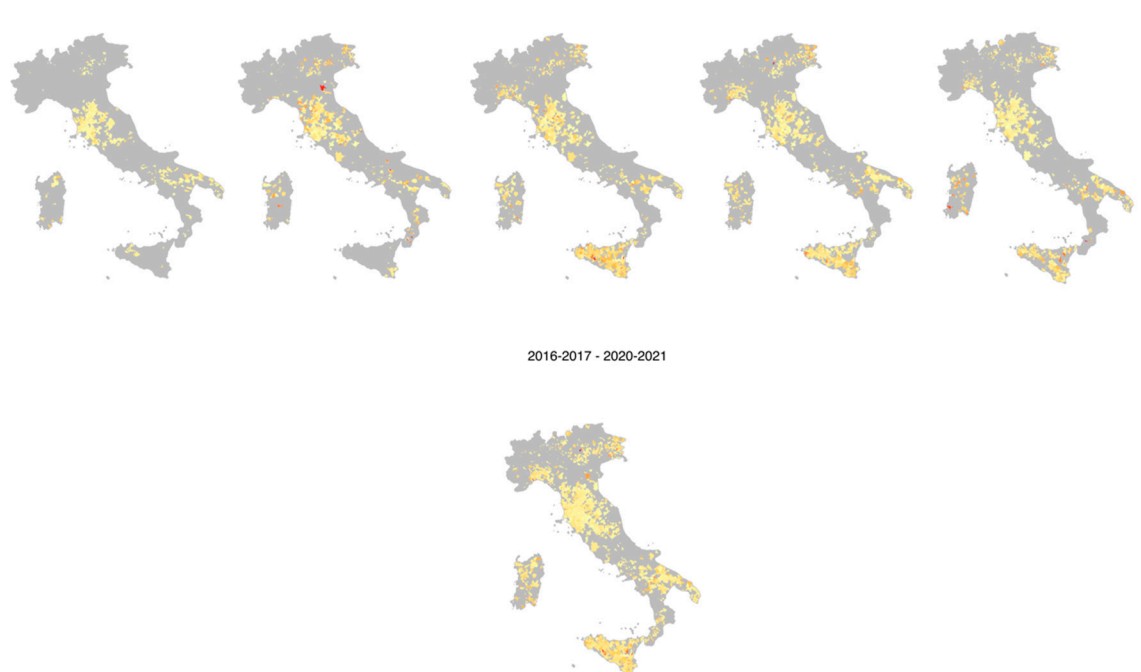

**Figure 1.** Spatial distribution of Woodcock abundance (Woodcock ICA) by Italian commune, by season (**top** row), and the average (2016–2017 to 2020–2021; **bottom** row). The warmer colours indicate higher abundance. Grey corresponds to non-available data.

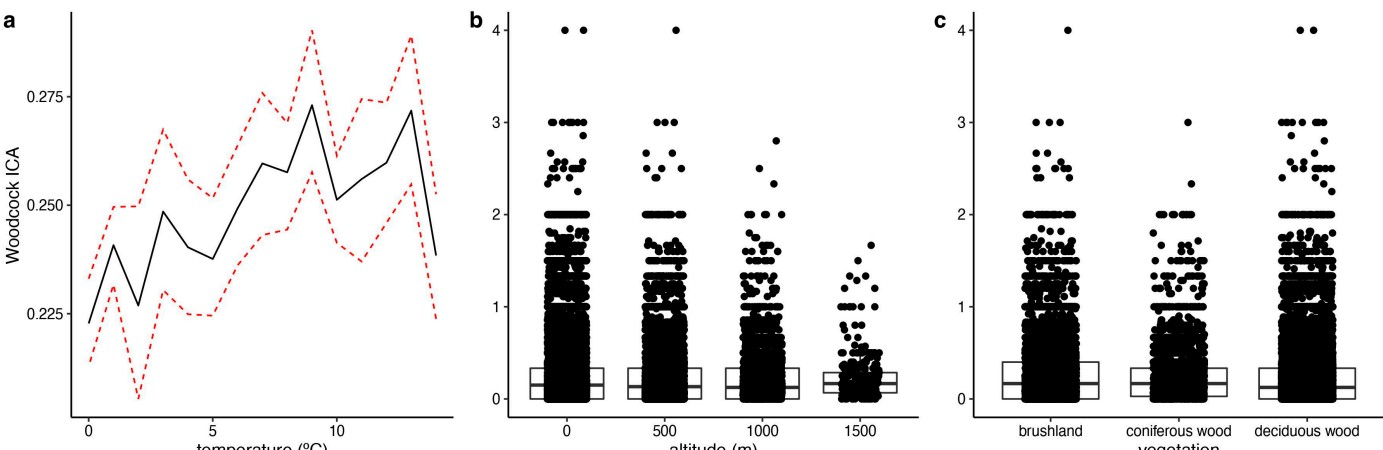

**Figure 2.** Variation of mean Woodcock abundance (Woodcock ICA, solid line) by temperature (**a**), dashed red lines correspond to 95% confidence intervals (95% CI) altitude (**b**) and habitat type (**c**).

**Table 2.** Estimates derived from the generalized additive mixed model (GAMM) for the effects of season, day in season, geographic location, and hunter in the number of Woodcock flushed.

|  | **Edf** | **Ref.df** | **Chi-Square** | *p*-**Value** |
|---|---|---|---|---|
| Season | 3.921 | 3.994 | 240.6 | <0.001 |
| Day | 7.920 | 8.000 | 6593.7 | <0.001 |
| x,y | 16.805 | 18.284 | 496.5 | <0.001 |
| Hunter | 938.483 | 1263.000 | 13,217.0 | <0.001 |

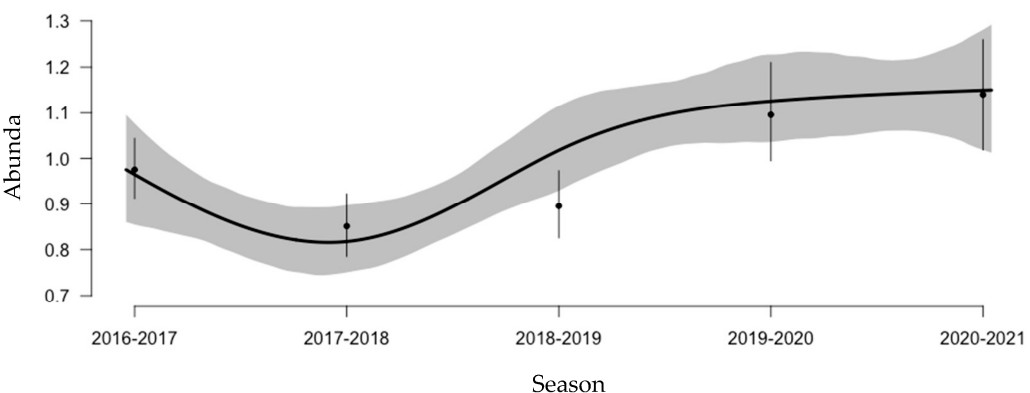

**Figure 3.** Variation of Woodcock abundance between seasons (2016–2017 to 2020–2021), fitted with a generalized additive mixed model. The grey area corresponds to the 95% confidence interval (95% CI). The points show the random effect of the seasons and the vertical lines correspond to their respective 95% confidence intervals (95% CI).

Finally, we considered the number of Woodcock flushed during monitoring trips across each year to evaluate the trend of Woodcock abundance in both, hunting and post-hunting, periods.

The abundance of Woodcock increased 4.75-fold from late September until around decades nine and ten of the hunting period, which correspond to late November and the beginning of December, and it suffers a 0.84-fold decrease until the end of the hunting period, in late January. During the post-hunting period, the abundance suffers a slight increase; in late March, the Woodcock is still present in Italy, at a level of abundance 1.17-fold higher than in early February, right after the end of the hunting period (Table 3; Figure 4).

**Table 3.** Result of generalized additive mixed models (GAMM) for the number of Woodcock flushed in Italy, between 2016 and 2021, during hunting trips undertaken from September to January, and post-hunting (February and March) trips with pointing dogs. The deviance explained by each model is shown in brackets.

| | Hunting Period (37.7%) | | | | Post-Hunting Period (31.1%) | | | |
|---|---|---|---|---|---|---|---|---|
| | Edf | Ref.df | Chi sq. | *p*-Value | Edf | Ref.df | Chi sq. | *p*-Value |
| Season | 3.97 | 3.99 | 554.8 | <0.001 | 3.54 | 3.86 | 14.3 | <0.001 |
| Decade | 2.99 | 3.00 | 598.4 | <0.001 | 1.58 | 1.82 | 63.1 | <0.001 |
| Commune | 401.62 | 1542.0 | 29,742.9 | <0.001 | 100.80 | 644.00 | 711.9 | 0.001 |
| Hunter | 769.51 | 1058.0 | 52,924.6 | <0.001 | 417.60 | 742.00 | 2749.9 | <0.001 |

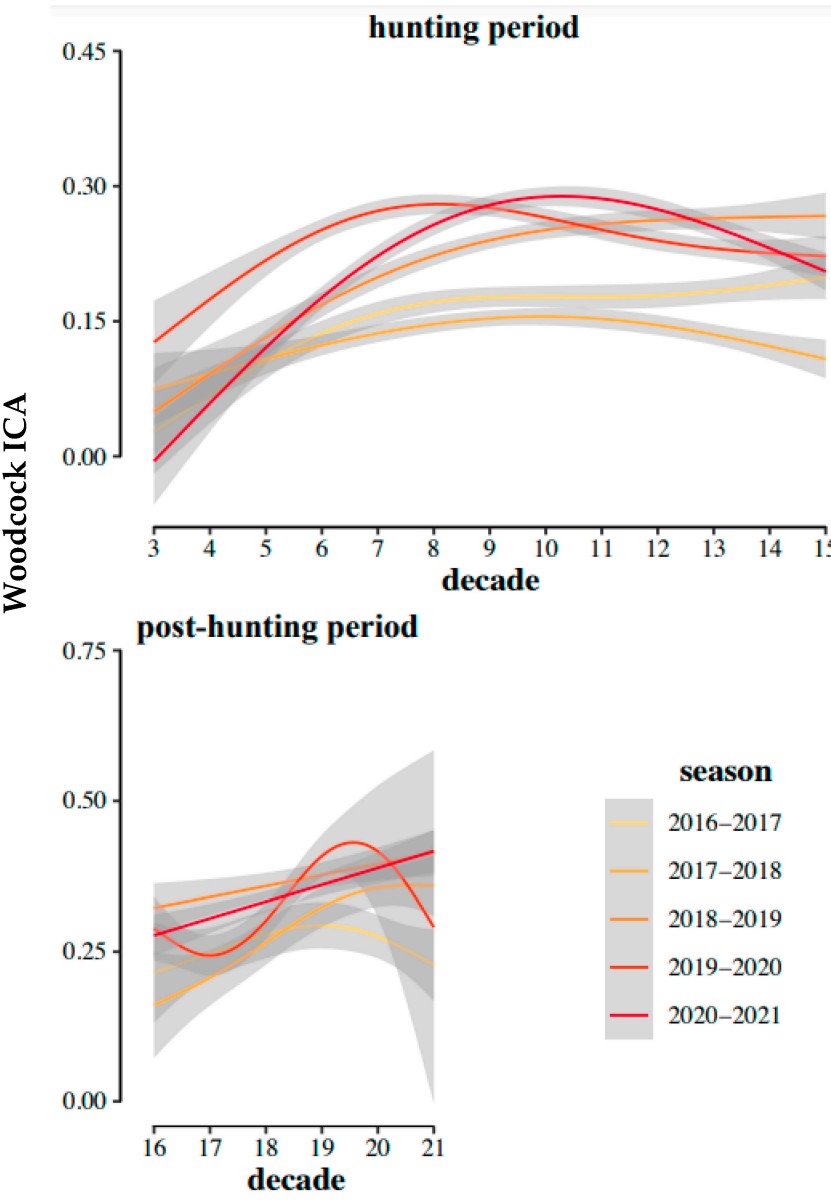

**Figure 4.** Variation in Woodcock abundance (Woodcock ICA) for each decade (decade 1: 1–10 September, and so on), in both the hunting and post-hunting period, for different seasons (2016–2017 to 2020–2021). The grey areas correspond to 95% confidence intervals (95% CI).

## 4. Discussion

In this study, our main objective was to investigate the abundance and distribution of Woodcock in Italy by comparing their presence during two sub-periods: from the 3rd Sunday of September to 31 January, and from 1 February to 31 March, each year. To achieve this, we utilized abundance indices which have been previously used in several studies on the migratory phases of Woodcock, particularly in their wintering areas [8,35]. The data collected here is essential for gaining valuable insights into the Woodcock species ecology and making informed management decisions for its conservation and sustainable harvesting. Nonetheless, it is important to note that this monitoring program may be affected by biases arising from non-stratified and/or random sampling methods and collaborations with non-technical personnel, including hunters [10]. However, the methodology adopted to flush the birds, namely the use of pointing dogs, and/or the data of hunting trips is well established to evaluate the abundance of several species among which the Woodcock is included [13,27,36]. The estimates we obtained in our study show a similar trend to those observed in independent data, which were obtained during night trips to capture Woodcock presence [27]. Additionally, our results regarding the variation within and between seasons align with findings from other studies in European countries with similar data [13,36].

Our findings indicate that there is a variation in the presence of Woodcock depending on the monitoring location, particularly when using the coordinates. Latitude, which is a useful proxy for factors with biological significance for organisms [37], has been shown to positively influence the densities of many resident bird species [38]. However, it may also have a significant impact on the presence of migratory species such as the Woodcock. Since we show similar abundance index values across the northern and southern regions of Italy, the abundance discrepancies at different geographic coordinates could reflect different migration timeframes in different Italian regions, even if a latitudinal gradient is not confirmed in this study. This could suggest that migratory pathways run northeast/southwest, with the Italian peninsula positioned obliquely in relation to these patterns [21,39]. On the other hand, the Carpathian Mountains could represent a barrier for the migration of distinct populations. Woodcock belonging to a Fennoscandian meta-population would migrate north of the Carpathians, while Woodcock belonging to a Mediterranean meta-population would migrate south of the Carpathians [7]. This variability is also confirmed by the morphological difference that emerged between Woodcock collected in the northern areas and individuals collected in the southern areas during the hunting period [14]: the northern woodcock was shown to have lighter plumage and greater body measurements, as pointed out also by Boidot [40].

Our findings suggest that temperature has a significant impact on the presence of Woodcock, with their abundance increasing in direct proportion to the occurrence of milder winter temperatures. These results are consistent with a study conducted in France over 14 consecutive winters, from 1984/85 to 1997/98 [41]. A species that makes extensive use of wet and muddy areas may have different opportunities to find trophic niches as a result of how the winter temperatures affected the soil's properties, i.e. relative hardness of the ground. The Woodcock selects habitats based on these properties, which are typically influenced by rain and other meteorological variables [41–43].

In Italy, the trend of annual and seasonal average cumulative rainfall has not changed significantly over the past 60 years (1961–2020), but if we only consider the last 35 years (1985–2020), the annual and seasonal averages have increased steadily [44], primarily as a result of exceptional and sporadic meteorological events. According to this trend, rainfall increases in the winter and autumn, especially in the peninsular regions [45]. This could increase the availability of resources and thus, Woodcock presence.

This scenario might have enhanced the resources available to Woodcock, which would have increased population abundance and length of stay in the Mediterranean basin. At the same time, the milder winters have made the alpine and pre-alpine areas favourable to

wintering of the species, where in the past it was temporally present only during migration periods.

In addition to climate factors, between 1985 and 2015, Italy's forests grew by 28%, going from 8,675,100 hectares to 11,110,315 hectares. Forests gradually and consistently expand until the percentage of land covered by woodland has reached 38% [46], increasing the habitat available for Woodcock. Wooded hedgerows and coniferous woodlands represent an important habitat for wintering, especially during the daytime, because this habitat type showed high availability of refuge areas [43,47,48]. Even though during the night-time Woodcock preferred meadows for a higher presence of earthworms, an important component of Woodcock's diet [43,47,49,50].

The number of Woodcock seen during the post-hunting period has steadily increased, and in the last three years, they have been comparable to those seen during the hunting period.

We obtained a yearly ICA value, higher than other areas of the Mediterranean basin, as north Spain [8], and similar to areas of France [27,51], but lower than other areas as south Spain, Switzerland, and Portugal [51], and our data confirm that Italy is an important area for the wintering phase of woodcock.

With respect to the trend of abundance per decade, the positive values in the second decade of September could testify to the presence of breeding contingents, especially in the Alps, confirming the presence of Woodcock in early September as reported by Spina and Volponi [39].

Our results show that Woodcock abundance increases significantly from the third decade of October to the second of November, which corresponds to the period of maximum post-nuptial migratory flux. This trend continues, albeit to a lesser extent, until the second decade of December, which defines the autumn migration period from mid-October to mid-December. However, the mean trend shows a continuous decrease until the end of January, with single seasonal curves showing fluctuations that vary from season to season, likely due to the meteorological trend. It was observed that some populations move south-southwest later and only if these countries experience cold snaps where frost becomes prevalent, for example, Balkan countries [52]. Similar behaviour was observed during winter, both in France, with more frequency, and in Spain less so, where some Woodcock (up to ca. 5%) change wintering sites even late into the season [53]. These movements can be considered erratic movements, and we think that in Italy this behaviour could be facilitated by the morphology of the landscape (i.e., proximity between mountainous and coastal areas) and due to the Adriatic Sea, that separates Italy from the Balkan Region by several kilometres. Furthermore, in particular conditions, Woodcock concentrate on certain biotopes, making altitudinal shifts, changing exposure, or reaching riparian or coastal areas. These movements can affect the value of the ICA if the hunting effort is also concentrated in these places.

The ICA index indicated an increased presence trend during the post-hunting season, up until the third decade of March, or with oscillations that led to larger values being recorded in the middle decades of the period than in the beginning ones. In all monitoring years, Woodcock registered ICA values for the month of March that were greater than those for the months of the wintering phase. This higher Woodcock concentration may result from opportunistic movements linked to climatic conditions, which are well-documented in other Mediterranean nations [36,54] but may be linked to migratory patterns. This specific, poorly understood component of the phenology of pre-nuptial migration will need to be clarified in future research and requires more in-depth analyses than is possible here.

The findings of our study appear to support previous research that the pre-nuptial migration toward nesting sites occurs over a shorter period of time, primarily between the second ten days of February and the third ten days of March (on average 40 days), as opposed to the autumn-winter migration, which occurs over a relatively long period of time (on average 70 days)., as evidenced by recent studies [21,36], and this supports our data derived from the post-hunting period, shorter than hunting ones.

A study based on ringing activity in Italy, showed a low number of individuals started to move from south to north, more than 100 km, in early January, however half of the sample started between the first and second decade of February [55]. It is therefore probable that woodcock mobility in January is attributable to momentary changes in the wintering site, due to the changing of weather conditions; this behavior has been well documented in Spain, where in January and in February the 64.3% of the woodcocks observed move from their original place, to go back there after just a week (pendulum movements) [54].

Our findings are consistent with this trend and support the hypothesis that the pre-nuptial migration starts in late winter-early spring in Italy when there is an increase in woodcock encounters during monitoring sessions.

## 5. Conclusions

The abundance of Woodcock in each decade, into which the annual monitoring periods were divided, allowed for the identification of three phases: the first phase corresponded to the arrival of migrating individuals in the autumn period (post-nuptial migration around the first ten days of October), the wintering phase, and finally, a phase of likely pre-nuptial migration beginning (starting from the second ten days of February).

Conditions such as a milder climate, increased fall and winter precipitation, and consistent afforestation have favoured the presence of Woodcock in Italy, for the wintering period, even where in the past was present only during the migration phases. In 2001, Woodcock presence was recorded in the Alps and northern Apennine only during the migration period, while in the Mediterranean wood, western side of Tuscany, it was observed for all wintering seasons. At the same time, observations at high altitudes (over 1.500 m) were rare and recorded woodcock only during migratory stopovers [56].

This phenomenon makes it more difficult to distinguish between migration and wintering behaviour, which is characterized by unpredictable and/or irregular movements. Furthermore, climate change in the western Palearctic may cause a delay in the autumn migration of some species of woodland birds to their wintering sites, complicating the description of migratory phenomena [57]. Our study was also affected by a climatic occurrence in the 2017–2018 season, which included a severe drought that lasted until autumn [58] and decreased the number of monitored Woodcock.

A longer monitoring period will be necessary to outline a more accurate trend, though our results give an important baseline to investigate the phenology of migration in Woodcock more granularly.

Some opportunistic movements are still possible during the wintering period, depending on the meteorological conditions, and this could be generating variation in Woodcock presence in different regions of Italy, perhaps replicated in other Mediterranean regions.

In terms of conservation, it is encouraging to note that the Woodcock maintains a steady presence in Italy throughout the winter season. Monitoring in all regions has confirmed their presence until the beginning of spring, thereby raising doubts about the exact timing of their pre-nuptial transit in the Mediterranean region.

Our study emphasizes the significance of establishing long-term monitoring programs for evaluating key spatial population metrics of game species, such as their presence and abundance within designated hunting areas, as well as outside of such areas for comparison. Such studies are essential for addressing conservation management and ensuring sustainable hunting practices. Our findings highlight the importance of such monitoring programs and the need for their continued implementation to support the effective conservation and management of game species populations. Finally, as the viability of game species populations is often modulated by abiotic and biotic factors, typically climate, resource availability, density-dependent effects, as well as hunting pressure, predator-prey interactions, and human disturbance, it is imperative that wildlife managers utilise monitoring data such as the data we collect here to disentangle the drivers of population fluctuations in a rapidly changing world. With this strategy, evidence-based management methods might be created to guarantee robust and long-lasting populations of game species [59].

**Author Contributions:** Conceptualization, M.T., P.P. and V.M.; methodology, M.T., P.B. and C.S.; validation, P.B. and T.M.R.; formal analysis, P.B. and T.M.R.; investigation, M.T., P.P. and V.M., resources, P.P., V.M. and C.S.; data curation, T.M.R.; writing-original draft preparation, M.T., T.M.R., K.J.M., P.B. and C.S.; writing-review and editing, all authors; visualization, K.J.M., T.M.R., P.B., P.P., V.M. and C.S.; supervision, T.M.R. and K.J.M.; project administration, P.P. and V.M. All authors have read and agreed to the published version of the manuscript.

**Funding:** This research received no external funding.

**Institutional Review Board Statement:** Not applicable.

**Data Availability Statement:** Data are available from the corresponding author on reasonable request.

**Acknowledgments:** We are grateful to the Woodcock hunters who participated in this monitoring project. We also want to express our gratitude to their pointing dogs, who spent a lot of time searching for the Woodcocks; without them, many of the Woodcocks would not have been identified. Also acknowledged is FIBEC—Federazione Italiana Beccaciai—(the Italian Woodcock Hunters Federation), for its assistance in fostering respect for this species and advancing knowledge of Woodcock.

**Conflicts of Interest:** The authors declare no conflict of interest.

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
