# Peer review of "Monitoring Eurasian Woodcock (Scolopax rusticola) with Pointing Dogs in Italy to Inform Evidence-Based Management of a Migratory Game Species"

_diversity, doi:10.3390/d15050598_

Round 1
Reviewer 1 Report
GENERAL COMMENTS
This paper presents the results of a survey carried out to assess the abundance and timing of the prenuptial migration of the woodcock in Italy. Even if it is an interesting topic, it is not original, it doesn’t present new methodologies and it doesn’t add new information to the subject area.
The text is not clear or easy to read. Concepts are non expressed in a logical sequence, some sentences seem to be off-topic, a number of relevant references are ignored and key information is omitted in the description of methods. Furthermore, grammar is not totally pure. See my specific comments for details.
As far as the adopted methodology, it does not seem to produce a reliable abundance index, since it does not account for multiple counts of a single individual during hunting and monitoring trips.
Finally, the conclusions drawn by the authors do not appear to be justified by the results presented in the paper.
SPECIFIC COMMENTS
Introduction
LINE 41: The correct citation of the birds Directive is "Directive 2009/147/EC".
LINES 53-54: it is not clear why you cited only some studies analysing age and sex ratio. There were other studies published in peer reviewed journals and carried out in Italy addressing winter mortality (Aradis et al 2008), diet (Aradis et al 2019), genetics (Burlando et al 1996, E. Trucchi et al 2011) and migration (Tedeschi et al 2020).
Aradis A., Miller M.W., Landucci G., Ruda P., Taddei S., Spina F. 2008. Winter survival of Eurasian woodcock Scolopax rusticolain central Italy, Wildlife Biology 14(1): 36-43
Aradis A., Lo Verde G., Massa B. 2019. Importance of millipedes (Diplopoda) in the autumn-winter diet of Scolopax rusticola, The European Zoological Journal 86: 452-457
Burlando B., Arillo A., Spanò S. 1996. A study of the genetic variability in populations of the European woodcock (Scolopax rusticola) by random amplification of polymorphic DNA. Italian Journal Zoology, 63: 31–36
Tedeschi, M. Sorrenti, M. Bottazzo, M. Spagnesi, I. Telletxea, R. Ibàñez, N. Tormen, F. De Pascalis, L. Guidolin, D. Rubolini 2020. Interindividual variation and consistency of migratory behavior in the Eurasian woodcock. Current Zoology66 (2): 155–163
E. Trucchi, G. Allegrucci, G. Riccarducci, A. Aradis, F. Spina, V. Sbordoni 2011. A genetic characterization of European Woodcock (Scolopax rusticola, Charadriidae, Charadriiformes) overwintering in Italy. Italian Journal of Zoology 78: 146-156
LINES 54-56: It is not clear what do you mean with “both methods, directly and indirectly”.
LINES 58-59: this sentence is badly written.
LINES 60-63: you should explain the difference between “abundance” and “occurrence” index.
LINES 67-68: it is not clear what is the reason why “in this context Woodcock monitoring programs are even more important”. In which contexts are monitoring programs less important?
LINES 67-69: how does “species richness” fit in this sentences? Your study is aimed at determining the abundance of a single specie. Species richness is a parameter to describe bird communities, not single species.
LINES 70-74: it is not logic to insert a sentence on the legal status of the woodcock between conservation status of the species and population estimates. Furthermore, I suggest to move this part of the text to the lines 50-51 (after the description of the range of occurrence).
LINES 78-79: the first decade of September does not correspond to the beginning of the wintering season.
LINES 79-80: the end of March does not “represent the last period of stay in the Mediterranean region”. Some individuals are still observed in April (see Tedeschi et al 2020).
LINES 81-81: it does not seem that this study “was carried out with the same […] sample effort in the Italian regions”. This statement is not supported by the description of methods and it seems to be contradicted by Figure 1.
Methods
LINE 92: in some regions the closing date of the hunting season was advanced. You should detail it in the text and consider the different closing dates when analysing your data.
LINE 95: it is not clear how you were able to estimate the number of different birds from the number of flushed birds. The experience gained trough the ringing activity reveals that during a ringing trip a single woodcock can be caught more times. The only way we have to avoid double counts is to check the ring. In the case of hunting trips, part of the woodcock is killed and this circumstance reduces the risks of multiple counts. However, the problem persists when the flushed birds are not hit or when monitoring trips are carried out outside the hunting season.
I believe that multiple counts may undermine your results.
LINES 98-98: you considered very few generic habitat categories. I think that it keeps you from obtaining more significant results.
LINES 125-127: it is not clear what this sentence means. What are the data not included in the analysis? What do you mean with “further analysis”?
Results
LINES 148-150: the lack of a latitudinal gradient in the abundance index could be the result of the methodology you adopted to analyse your data. You should give more details on your analysis.
LINES 149-143: the absolute number of hunters and dogs involved in monitoring and the mean number of hunters and dogs for monitoring trips should be specified.
LINE 154: I suggest to make more evident the differences among abundance classes.
LINES 166-168: it is not clear what is represented in Figure 2, graphs b) and c). Is there an overlap of dot-plots and box-plots? Furthermore, some dots aligned horizontally seem to be the result of a methodological error. The legend of the axes should be changed. On the x-axis I suggest to write “abundance index”. On the y-axis the text must be translated into English.
LINES 177-179: the comparison among years is strongly biased by the uneven monitoring effort across the study period. As it can be deduced from Figure 1, in the seasons 2016-17 and 2017-18 very few monitoring trips were carried out in Sicily, a region where a high abundance of woodcock was recorded. That could likely influence your results. You cannot obtain reliable trends if you compare inhomogeneous data.
LINES 190-195: it is reasonable to assume that the phenology of the woodcock is not the same across Italy, being site-specific. For instance, I expect strong differences between north eastern regions and Sicily. If you did not follow a standardized monitoring protocol, as you explicitly admit (LINES 213-214), your results could be influenced by the uneven distribution of monitoring efforts across regions.
Discussion
LINES 212-214: it is not clear what do you mean with “crucial information for the understanding of the species and the management options to be implemented”. If you are referring to the onset of the pre-nuptial migration, your conclusions are not supported by your data (see below).
LINES 263-265: this sentence is not supported by data. Early arrivals of foreign woodcock are known to occur in mid September (see Spina and Volponi 2008).
Spina F., Volponi S. 2008. Atlante della Migrazione degli Uccelli in Italia. non-Passeriformi. Ministero dell’Ambiente e della Tutela del Territorio e del Mare, Istituto Superiore per la Protezione e la Ricerca Ambientale (ISPRA). Tipografia CSR-Roma.
LINES 270-273: the year-by-year approach you followed to describe the temporal variation of the abundance index in the annual cycle does non allow to drawn any conclusion on the onset of the pre-nuptial migration, given the high inter-annual variability.
LINES 203-205: winter “fluctuations which vary from season to season” are hardly related “to the meteorological trend”. The winter movements induced by a worsening of weather conditions and described for woodcock wintering in France (“escape migration”) have not been confirmed in the case of Italy. Furthermore, this winter displacement of birds occurs only occasionally during very cold weather conditions: “Escape migration to Spain was a rare behavior estimated to occur in less than 1% of individuals during ‘normal’ years. […] However, a notable increase in escape migration probability (up to ca. 5%) occurred during, and supposedly as a consequence of, the high intensity cold spell” (Péron et al 2011). Inter-annual variations of the abundance index are more likely due to the uneven distribution of monitoring efforts across Italy (see above).
LINES 282-282: I do not understand where you see the "bell" that starts in the second decade of February and reaches a peak in the second decade of March. Have you considered only the trend of the season 2019-20 in Fig. 4? If so, it is not clear why you considered that season the most representative.
LINES 282-284: The higher average value of the abundance index in the monitoring period compared to the hunting season can be explained by the higher probability of multiple counts during monitoring trips without shooting (see my comments referred to LINE 95).
LINE 288: when discussing the timing of the pre-nuptial migration you should consider the following key publication:
Bairlein F., Mattig F. Ambrosini, R. 2022. Analysis of the current migration seasons of hunted species as of Key Concepts of article 7(4) of Directive 79/409/EEC. In: Spina F., Baillie S.R., Bairlein F., Fiedler W., Thorup K. (eds) The Eurasian African Bird Migration Atlas, EURING/CMS. https://migrationatlas.org.
Conclusions
LINES 294-296: the explanations for the supposed increase of the number of woodcock in Italy are not adequately supported by evidences. Your hypothesis that this trend is linked to “milder climate and an increase of precipitation especially in autumn and winter season” should be supported with ad hoc references. On the other hand, the expansion of forest surface is a very slow process whose effects cannot be evident during a study period of just 5 years.
Author Response
Please see the attachement below
Regards,
Paolo

Reviewer 2 Report
I have carefully read the ms. The subject is interesting concern the abundance of Woodcock in two years in Italy. However, lack of details leads it is difficult to evaluate the merit in the ms.
1) The rationality of the abundance indices, used in the ms, should be explained in detail. "The abundance indices have already been used in other studies relating to the migration phases of the Woodcock". The authors need to provide the reasons why the previous research and this study use this abundance index. Is the same abundance index used in this study and previous research [29, 8]?
According to line 110, it seems "number of different Woodcocks contacted" is linearly related to the trip duration, number of participants, number of pointing dogs used. This linearly relationship should be proved before the author using the formula (line 110).
2) More details concern data collection should be provided. Which methods were utilized to control the variation of investigation intensity in different areas? Did the survey be conducted in remote places?
3) The discussion can be improved in two ways: compare the Woodcock abundance with other winter birds abundance in the study area; compare the Woodcock abundance in the study area with Woodcock abundance in other research.
4) Too many short paragraph. Most short paragraphs need to be merged.
Author Response
Please see the attachment below
Regards
Paolo

Reviewer 3 Report
Woodcock paper
Introduction should be rewritten and re-organized. Start with the role that monitoring programs play in helping determine conservation status and harvest regulations for birds. Then explain how monitoring schemes are especially needed for cryptic species, which require specially designed survey methods to generate useful indices of abundance. Next, explain that data gathered by hunters can be a way to augment data from standardized survey methods and may provide the only source of data for some species in portions of their geographic range.
Next, introduce what is currently known about woodcock distribution and abundance and introduce enough of the biology to explain that it is very difficult to survey well. Then you can introduce the information gaps related to Italian populations and the opportunity to work with hunters to fill those gaps. At the end of that paragraph, clearly state your main objectives.
I think you can write a solid introduction in just 2 paragraphs. Nothing more is needed. Note that your current introduction is simply many sentences set apart as paragraphs. It is better to organize all your thoughts in a logical sequence and have 2 paragraphs to introduce your work.
Methods
I think you should briefly summarize the key elements of the training course the hunters took. What were they taught that is directly relevant to your study?
Line 106—explain the difference between a standard duration trip and the “effective duration”
I ask because I do not understand the equation for ICA and why it is multiplied by 3.5 when trip duration is already in the equation. The way the equation is currently presented seems to indicate 4 division operations but the sequence these occur in is unclear. Please clarify how this works better.
Alternatively, can you simply say that you are calculating an encounter rate? This would be:
Number of woodcocks encountered per hour (or other standard time amount)
Divided by
The total number of hunters plus dogs
I don’t think you would need to multiply by 3.5 then as long as you had realistic bounds around the minimum and maximum amounts of efforts you include. That is, don’t accept any hunting data from effort of, say, less than one hour, or more than, say, 6 hours. This reduces the chance that outliers (very short or very long trips) will influence the distribution of the encounter rate.
Even with that method (and with any method you might use), it will be important to state explicitly that you are assuming equal skills among observers at counting and identifying woodcocks.
Stats
Because the introduction does not really yet clearly communicate the primary objectives, I had a hard time making sure the stats link clearly to your goals. I recommend revising the stats section to more clearly state what exactly you are trying to do with each statistical approach. It’s best not to ask too much of readers to try and guess your goals. And it’s best if your work is clearly reproducible by others, which I don’t think it currently is because the organization of the stats section is not transparent enough at the moment.
Results
Why more trips than woodcock hunting trips? If this paper is only about woodcocks, then perhaps it is better to just state the number of woodcock hunting trips.
By the way, that is a lot of hunting trips!
Because I did not really understand the ICA based on its presentation, I have a hard time evaluating the results. They may change if you adjust to encounter rate instead. So I will comment here on the general pattern in which results are presented.
How was the primary habitat information of each survey measured? I don’t remember seeing any methodological details about that.
When was temperature measured? Beginning of each survey, middle or at the end?
Figs 2 b and c, there are so many data points that the patterns are perhaps too obscured. Could you hide the individual data points and just do a typical box and whisker plot with points for just the extreme outliers (e.g., 3 SD and greater?)
Line 181, I’m not sure your use of ‘decades’ is typical. Can you find an alternative word to explain what you mean here? Do you just mean ‘week’?
Discussion and other thoughts
To be honest, the difficulties of how the English is framed limits my ability to understand exactly what the authors are concluding. I recommend finding a native English speaker as a colleague who can truly help you clarify the language, which will then facilitate the communication of your findings and what they mean.
It seems to me that a great Discussion would emphasize what you have learned about the abundance and phenology of woodcocks in Italy and how those patterns are the same or different from other similar studies in different regions, how the availability of data from hunters more generally is useful for learning about cryptic species and helping inform conservation and management plans, what concerns or possible shortcomings are present in your data given that no data are perfect, and what the remaining knowledge gaps are.
I enjoyed the reading the paper and appreciate your work. I do encourage a better-explained flow of information and logic so that readers can appreciate and evaluate your study more completely.
Author Response

(The authors gave the same response as above.)

Reviewer 4 Report
Wintering monitoring of Woodcock (Scolopax rusticola) on the hunting area in Italy
- A brief summary (one short paragraph) outlining the aim of the paper, its main contributions and strengths.
The article presents data on the abundance and distribution of woodcock Scolopax rusticola in Italy during wintering and spring migration. The data were collected based on monitoring counts conducted by hunters during 5 autumn-winter seasons, from 2016-2017 to 2020-2021. Factors affecting the distribution and abundance of woodcocks have been determined.
- General concept comments
Article: highlighting areas of weakness, the testability of the hypothesis, methodological inaccuracies, missing controls, etc.
The article needs minor changes. The methodology is too briefly presented and needs to be developed and supplemented with some points.
In the Results section, use the indicators introduced in the Methodology. In the figures, the descriptions need to be revised and completed
General questions to help guide your review report for research articles:
- Is the manuscript clear, relevant for the field and presented in a well-structured manner?
Yes
- Are the cited references mostly recent publications (within the last 5 years) and relevant? Does it include an excessive number of self-citations?
Yes
- Is the manuscript scientifically sound and is the experimental design appropriate to test the hypothesis?
In the paper there are no hypothesis
- Are the manuscript’s results reproducible based on the details given in the methods section?
No, methods should be completed
- Are the figures/tables/images/schemes appropriate? Do they properly show the data? Are they easy to interpret and understand? Is the data interpreted appropriately and consistently throughout the manuscript? Please include details regarding the statistical analysis or data acquired from specific databases.
Descriptions should be corrected on the figures. Statistical methods are correct.
- Are the conclusions consistent with the evidence and arguments presented?
Yes
- Please evaluate the ethics statements and data availability statements to ensure they are adequate.
They are adequate.

Author Response
Please see the attachment
Regards
Paolo

Round 2
Reviewer 1 Report
Authors replied properly to some comments, but avoided to correct the manuscript in a substantial way, following my previous criticism.
Below the main issues still controversial in the revised version.
It is not enough to describe in Methods the limits of the methodology adopted to contact the birds. Data should be critically presented and discussed throughout the manuscript taking into account the potential bias originated by the weakness of the method. Authors should also be more cautious in interpreting their results.
The conclusions drawn by the Authors do not appear to be justified yet by the results presented in the paper. Particularly questionable are the sentences on the onset of prenuptial migration, that have relevant implications for the determination of the closing date of the hunting season.
At this regard, the sentence in LINES 372-375 is completely misleading. We cannot deduce any sound conclusion from Fig 4, because woodcock abundance is reported separately for the periods September-January and February-March. To determine when the abundance index increases (revealing the onset of the migration) we should have a single graph for the whole study period (September-March), but the adopted method does not allow to produce a single graph. Once again the method used to calculate the abundance index shows its limits.
Author Response
Please see the attachment
Regards,
Paolo Bongi

Reviewer 3 Report
It seems the authors have taken many steps to improve the paper. I think there may still be areas of confusion, some of which can be improved with more editing of the English. Others involve a persistent lack of clarity about exactly how the data were organized and analyzed. Some of the worries about such things could be assuaged if you archive the data files you used so that others interested in woodcocks could re-analyze the data with other approaches if they so desire.
Author Response

(The authors gave the same response as above.)

Round 3
Reviewer 1 Report
The manuscript has been improved compared to the initial version. However, the Authors’ replies did not fully resolve the issues I previously raised and the text is still unsuitable for publication.
There is much room to improve the discussion, avoiding comments not supported by data originating from the study. As I already pointed out, Authors should be more cautious in interpreting their results, taking into account my previous comments.
Regarding the onset of pre-nuptial migration, the new wording (LINES 401-422) responds to my comment only partially. In addition, many inconsistencies have been introduced in the new text.
LINES 401-403: this sentence is not clearly written. What it means “rising presence peaks”?
I think that it is incorrect to use “ICA” to indicate an index not derived from hunting. To avoid misunderstanding, different acronyms should be used throughout the text for indices derived from “hunting trips” (true ICA) and “monitoring trips”.
LINES 403-405: it is not correct to compare indices obtained with different methodologies. As I already remarked, the higher average value of the abundance index in the monitoring period compared to the hunting season can be explained by the higher probability of multiple counts during monitoring trips without shooting.
On the other hand, it is not clear why you write that ”the higher average value of abundance in post-hunting period compared to the hunting season it is due to a lower monitoring effort”. Was the monitoring effort in the post-hunting season too low to obtain a reliable index? If so, you should avoid to discuss data which are not sound enough.
LINES 407-412: which concerns are raised by your findings? It is not clear what do you mean. I do not see any result suggesting “unpredictable movements” of Woodcock. I propose to delete this sentence.
LINES 413-416: this sentence is completely wrong. The hunting and post-hunting periods do not correspond at all to the post-nuptial and pre-nuptial migration, respectively. You draw conclusions meaningless.
LINES 417-419: reference [54] does not give any evidence that early movements from south to north are “less than 100-km”. This reference, indeed, reveals that the pre-nuptial migration starts in January rather than in “late winter-early spring” as written in LINE 419.
LINES 419-422: your data do not allow either to confirm or contradict that pre-nuptial migration starts in January, as showed by reference [54]. I consider more appropriate to cite the reference [54] at LINE 318.
After reviewing the Discussion, other parts of the manuscript (Abstract, Introduction and Conclusion) should be modified accordingly. Below I indicate some of the changes that are needed.
LINE 25: not only “wintering phase and pre-nuptial migration phase” but also post-nuptial migration. You have more sound data, indeed, on post-nuptial migration than on pre-nuptial migration.
LINE 34: what are the “fresh perspectives for potential management strategies in the future” provided by your work? It is not clear. I suggest to delete this sentence.
LINE 96: you investigated also the post-nuptial migration.
LINES 108-111: this sentence is not clear.
LINES 424-427: this sentence is wrong. The “peaks of the curves of abundance per decade” do not represent the “migratory periods”, but the peaks of the migratory periods. Your data do not allow to identify when pre- and post-nuptial migrations start and end. You need to rephrase this sentence.
LINES 473-475: it is not clear the meaning of this sentence. I propose to delete it.
Finally, I suggest the manuscript be revised by an English native speaker.
Author Response
Please see the attachment.
Regards
Paolo
